# Thinking Outside the Box: Deepening Private Sector Investments in Fiji's Nationally Determined Contributions through Scenario Analysis

**Jale Samuwai [1,2,*]** , **Jeremy Maxwell Hills [2,3]** and **Evanthie Michalena [3]**

1   Poly Micro Cluster, Oxfam in the Pacific, Suva, Fiji
2   Institute of Marine Resources, The University of the South Pacific, Private Mail Bag, Suva, Fiji
3   Sustainability Research Center, The University of the Sunshine Coast,
    Maroochydore DC, QLD 4558, Australia
*   Correspondence: jalecuruki5@gmail.com

**Abstract:** Private finance is seen as the financing panacea for resourcing the nationally determined contributions (NDC) submitted by 170 countries to the United Nations (UN) system. Mobilizing private investment is challenging, especially for vulnerable Pacific Island Countries (PICs). Fifteen PICs have already submitted ambitious NDC targets, in which transition towards a sustainable energy environment through investment in renewable energy (RE) is central. Presently, RE investments in PICs are primarily external donor financed, however, reliance on limited and uncertain external finance is unlikely to deliver the required energy transition. A future scenario methodology was used, with Fiji as a case-study; the analysis provided insight into alternative trajectories towards transition. Based on the scenario analysis, an NDC resource mobilization framework was developed. Conclusions suggest that donors should re-orientate their priorities from investments in RE installations, towards investments that upgrade the current RE readiness levels and promote a long-term perspective of "organically growing" the local private RE sector. Channeling resources to target initiatives that will endogenously grow the domestic private sector is critical for PICs, as well as other developing countries, which represent a majority of the NDCs, and which are projected to dominate global growth in energy demand for decades to come.

**Keywords:** climate change; nationally determined contributions; renewable energy; electricity sector; climate finance; private sector; Pacific Island Countries

---

## 1. Introduction

"Why is it that private investment in renewable energy (RE) is not easily forthcoming for Pacific Island Countries (PICs)?". This is a lingering question that many PICs and development partners continue to grapple with, despite undertaking various initiatives in order to promote private finance investments in the energy sector. Investment in RE has been a standing priority for the Pacific region, because most PICs are heavily dependent on expensive fossil fuels. Threats of climate change have renewed the sense of urgency to scale up RE investments in PICs. RE is advocated as a global mitigation necessity, and also as an enabler of resilience for countries to climate change impacts.

The renewed motivation towards RE investments globally is, to a larger extent, driven by the 2015 Paris Agreement (hereon referred to as the Agreement). The Agreement aims to transition global economies towards a low carbon development pathway by limiting the rise of the global average temperature to below 2 °C above pre-industrial levels, and to encourage efforts to limit the increase to below 1.5 °C. Critical to the achievement of this goal are countries' nationally determined

contributions (NDC), which contain the pledges they have made in terms of emission reductions and resilient development [1].

Investment in RE is at the core of the NDCs submitted to the United Nations Framework Convention on Climate Change (UNFCCC) [2,3]. So far, 170 countries who are parties to the UNFCCC have submitted their NDC. Energy use and generation account for two thirds of the world's greenhouse gas emissions [4]. The transformation of the global energy sector through investments in RE is essential for achieving the Agreement's objective [3].

Public finance is the primary source for RE investment globally [5]. A huge public climate finance gap, however, exists, which hinders the global implementation of NDCs [6]. At least 1.5 trillion USD is needed every year to close the current investment gap [7]. The NDC financing pathway as per the Agreement is very vague and uncertain. While developed countries have committed to mobilizing 100 billion USD a year from public and private sources by 2020 [1], they, however, did not commit towards individual financial targets. Rather, developed countries will decide on a voluntary basis how much climate finance they will provide, as well as over what time period, in what form, and through which channels [8]. The uncertainty that surrounds external public climate finance undermines the abilities of developing countries, particularly the abilities of small and poor developing countries like the PICs, which are challenged with severe resource limitation and are heavily dependent on international climate finance to fulfill their obligations as per the Agreement [9,10]. Countries like the PICs must now rethink strategies on how to attract and mobilized new and innovative resources that will source sustainable finances to implement their NDC.

Private financing has been advocated by developed countries and development partners as the panacea for the shortfall and the uncertainty of public financing sources in developing countries [3,11]. Two major factors drive the focus on the private sector, as follows: (1) The private sector is the custodian of a large pool of capital that could be directed towards climate change activities [12]. It is estimated that market value of assets, corporate and government bonds, and loans that are managed by the global financial sector alone are worth USD 225 trillion [13]. Secondly, private finance has catalytic properties that could effectively scale-up the "reach" of public finances [11,13]. In the right environment, a given amount of public finance could leverage 3–15 times the amount of private finance [14]. However, the suitability and the success of strategies that stimulate private sector investments have been a "mixed bag" across developing countries, because of the heterogeneous nature of countries' climate change and economic context [15].

Using the case of Fiji, this paper explores a potential resource mobilization strategy, which could enable the domestic private sector to "deepen" its investment in Fiji's NDC. The NDC resourcing roadmap presented in this paper can serve as guidance to other Small Island Developing States (SIDS) on how best to use external public finance to leverage their domestic private finance effectively.

The structure of the paper is as follows; Section 2 provides an overview of Fiji's energy landscape. Section 3 discusses the method used to develop the proposed resourcing frameworks. Section 4 presents the results, followed by Section 5. Section 6 provides the concluding remarks and the way forward for future research.

## 2. Country Overview

### 2.1. Fiji's Energy Sector

Fiji is an archipelago of more than 300 islands, with a total population of 918,000. Fiji is vulnerable to sea level rise and natural disasters, which are made worse by climate change, such as cyclones, flooding, and drought.

Electricity is regarded as the low hanging fruit for low carbon transition in the Pacific [15]. Fiji's current energy mix consists of 53% hydro, 45.5% diesel and heavy fuel, 0.39% wind, with the remaining 1.1% supplied by independent power producers (IPPs) [16]. Grid-based electricity demand is concentrated in urban areas [17]. The current electrification rate is around 96% in urban and 82%

in rural areas [17]. Fiji's energy coverage is comparatively better considering the rest of PICs [16]. However, Fiji is heavily dependent on imported fossil fuel in order to sufficiently meet its electricity and overall energy needs [18], and does not possess any established oil reserves. Evidence indicates that Fiji's fuel imports accounts for 14%–17% of the gross domestic product (GDP), which is relatively higher than in other PICs [19,20]. Fiji's annual spending on fossil fuels is estimated to be 310 million USD per annum [17], of which 22% is dedicated to generating grid-based electricity [16].

The burdensome cost of imported oil threatens the overall achievement of Fiji's sustainable development goals, because significant national resources are diverted from critical development initiatives such as health, education, and infrastructure [18]. For Fiji, investing in RE is motivated by reasons that span economics, geopolitics, and health and livelihood resilience issues, with energy security and poverty alleviation being highlighted as the two key objectives [19,21]. RE in Fiji is both a mitigation and a resilience building initiative. RE is seen as critical in reducing Fiji's vulnerability to climate change, as well as Fiji's exposure to external market shocks [19].

### 2.2. Financing Fiji's NDC: The Road to 2030

Fiji has set an ambitious emission reduction trajectory of the electricity sector by 30% by 2030 [22]. A two-pronged approach is planned, where 10% will be through economy-wide investment in energy efficiency, and 20% will be achieved through a radical transformation of its current grid-based electricity sources to be 100% sourced from RE [22]. Of the 30% business as usual reduction, the government of Fiji (GoF) expects that 10% will be achieved unconditionally using domestic national resources, while 20% will be conditional on the receipt of significant means of implementation and support from other sources [22].

To fully implement its NDC by 2030, Fiji will need an estimated USD 2.95 billion [22]. The enormity of the scale of investment required for the NDC though, outpaces Fiji's current ability to finance the change envisioned. The GoF has also explicitly acknowledged that its economy is not adequately equipped to pursue expensive financial instruments, which will add to its current debt burden [22]. As a consequence, the GoF has conditioned the overall success of the NDC on the receipt of USD 1.67 billion from financing sourced outside of the national budget [22].

Recent RE studies in the PICs context, like those from the authors of [18,21,23], have made numerous policy suggestions on how to deepen the domestic private sector role in the energy sector. These policy suggestions tend to revolve around legislative and regulative reforms of the energy sector. However, none have actually explored or suggested a potential financing framework that is needed to "deepen" the domestic private sector investment in RE in PICs. This is the gap that this paper seeks to address.

### 3. Method

This study used the scenario methodology to develop the strategic options of RE investment for Fiji. This methodology is used to provide a more analytical basis for policy options backed up by documented information and expert perceptions. The scenario method focusses on significant but uncertain barriers for RE, for which strategic planning is vital. This method is different to just ranking the most significant barriers to RE based on literature or expert consultation. The developed options are based on a sequential methodology that uses published information, consultation with RE regional experts, and a future scenario approach to identify a number of emergent strategies for RE investment.

The scenario methodology is a strategic planning tool for improving decision making against the background of possible future environments [24]. Scenarios allow users to envision how possible futures might logically unfold by deciphering how current conditions in a specific environment might evolve [25]. Scenarios are most useful in situations where critical decisions about the future are to be made in an environment that is highly complex and dynamic [24], which mirrors the case of the Fiji RE investment.

Scenarios facilitate users to consider unexpected issues in the operating environment, allowing them to "think the unthinkable" by exploring new horizons, and consider an alternative future by challenging existing assumptions [24]. The scenario analysis technique has been pervasively used, and has been proven to be very successful in the area of strategic planning, especially in the field of corporate investments and defense (i.e., military intelligence); the global dominance and competitiveness of Shell in the global energy sector has been attributed to the use of scenario planning [26]. Within the context of resource mobilization, scenarios tend to be very effective in developing robust investment strategies against an uncertain future [26]. Unlike other planning tools, scenarios focus on the area of "critical uncertainty" in achieving an objective, and they systematically develop several plausible alternative environments in which the objective could be achieved [26]. By focusing on issues of critical uncertainties, scenarios allow users to examine issues that would not have be considered, and thus, they tend to be more effective in dealing with "big picture issues" and setting strategic directions, rather than short term technical decisions [26]. This structured approach to thinking about the future has enabled organizations to be strategic about where and how to direct resources in the mid- and long-term, as they try to secure viable and long-term success [26]. All of the above features make scenarios an appropriate part of a methodology in terms of RE investment in Fiji.

*3.1. Applying the Method*

This study used a five-step methodology from previously published scenario analyses by Blyth [24] and Gray [27].

3.1.1. Step 1: Identifying the Barriers to RE Investment

The authors conducted a thorough review of the literature to identify the barriers that have been consistently highlighted as critical inhibiters of investments in RE. The literature review covered all of the available RE information from scientifically published work, as well as grey material sourced from development partners. This pre-identification of barriers was used firstly to provide the broadest possible preview of RE barriers to initiate the scenario process, and secondly, to form a common starting point for subsequent engagement with experts, to allow for consultation information to be coherently collated. A total of 50 major investment barriers were identified from the literature review. These barriers were drawn across the spheres of politics, environment, social, economic, and technology.

3.1.2. Step 2: Ranking Barriers by Significance and Uncertainty

This step involved consultations with experts from the Pacific region. A total of 15 climate finance experts working in international, regional, and national agencies were interviewed by the lead author. These interviews with individuals were carried out when the Development Partners in Climate Change (DPCC) meetings were regularly convened by UNDP (Pacific Regional Office, Suva). This setting is a collaborative platform for discussion amongst climate finance experts of the GoF and its development partners (donors). Attendees to this meeting tend to be consistent, as the participating organizations usually send the same experts. Private sector experts, on the other hand, were drawn from financial institutions in Fiji. Those were interviewed separately during the DPCC meeting by the lead author. A total of five private sector experts agreed to participate for this study. Overall, a total of 20 experts participated in this study.

Initially, preliminary interviews were conducted with the experts to identify the most critical barriers from a list of 50 barriers, which were derived from the literature. The list of investment barriers was modified, with some barriers being merged and others being discarded based on expert perceptions; finally, 25 primary barriers emerged.

A Likert scale was then developed for the 25 barriers to allow experts to rank the level of significance and uncertainty of each barrier, ranging from zero (0, insignificant/highly certain) to five (5, highly significant/high uncertainty). At subsequent DPCC meetings and bi-lateral private sector interviews, experts ranked the 25 barriers by significance and uncertainty. During these consultations,

their own barriers were plotted on their respective axes, and experts where given a chance to review and confirm the information provided. At the end of this stage, Likert scores for 25 barriers by 20 experts were collected.

### 3.1.3. Step 3: Plotting the Barriers by Significance and Uncertainty

The results of Step 2 were then plotted onto axes of "significance" and "uncertainty" (Figure 1). The origin of both of the axes in the plot was the mid-value of the Likert scale, which splits the axes into four quadrants. The circles in the figure indicate a rounded envelope bounding the majority of the responses of the experts; this graphical approach is used in a scenario analysis to help keep figures visually clean but based on semi-quantitative scorings.

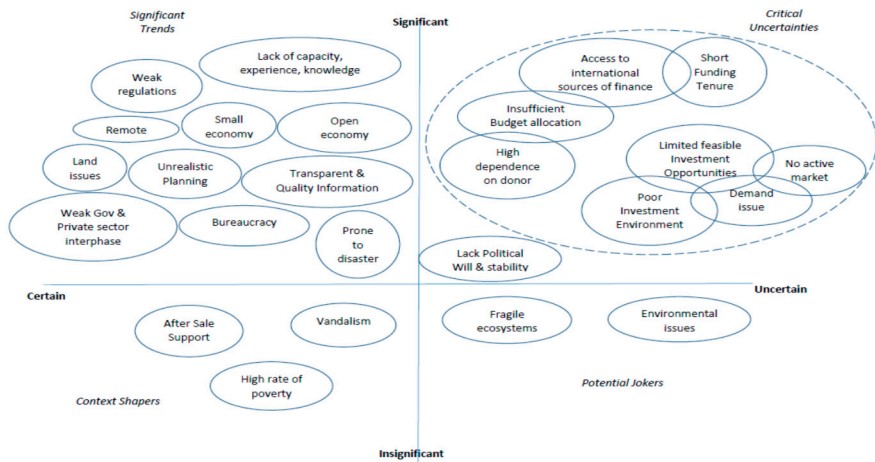

**Figure 1.** Barriers to renewable energy (RE) investments on axes of significance and uncertainty.

Barriers that are highly significant and uncertain are those that are unpredictable in nature and particularly important for Fiji. Barriers that fall inside the "significant" and the "certain" quadrant are classified as significant trends, and these are the predetermined barriers whose influence are more predictable and are expected to have a significant impact on the topic [24]. Those barriers that fall in the "low significant" and "certain" quadrant are characterized as context shapers, meaning that they are relatively certain, but tend to have an impact on the broader environment [24], and those barriers that fall in the "uncertainty" and "low significant quadrant" are classified as potential jokers, meaning that these are issues that are highly uncertain, but are not expected to have much impact on the topic [24].

### 3.1.4. Step 4: Creating New Emerging Axes

This step focuses on barriers that fall in the significant and uncertain quadrant, for which future planning is imperative. The barriers in this quadrant were iteratively clustered to form new axes of polarity, which best described these significant and uncertain barriers. Through trial and error by the authors, the two emergent clusters, which provided the most logical consistency of the representative barriers, were donor dependence, and investment environment and market. Only one barrier, the "lack of political will and stability", was not analyzed, because it is an issue outside the control of the internal RE private sector and is a fundamental prerequisite to any future progress in RE. The two emergent cluster areas were then extended into axes spanning low to high donor dependence, and low to high quality of investment environment and market (Figure 2).

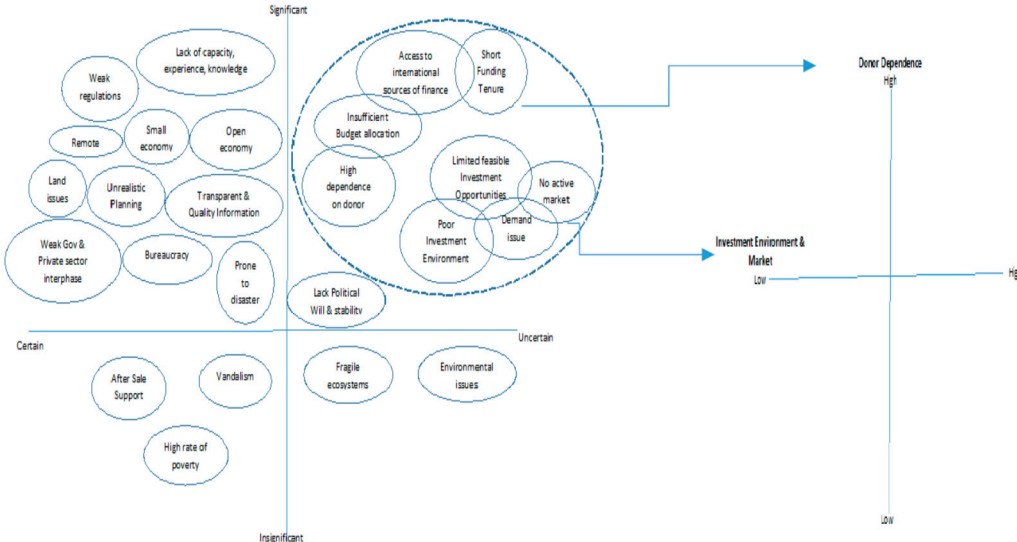

**Figure 2.** Creating new axes of polarity from the barriers in the significant and uncertain quadrant.

### 3.1.5. Step 5: Developing the Scenarios

The new axes, reflecting the combined significant but uncertain barriers, provide four new quadrants, each of which represents a possible scenario. Detailed scenarios were developed based on these four quadrants, which have varying levels of donor dependence and quality of investment and market. Following the method of Blyth [24] and Gray [27], four descriptive scenarios were developed, each reflecting a different combination of donor dependence and investment environment (see Figure 3). The scenario methodology forces a focus on particular significant but uncertain barriers, for which developing strategies and flexible planning would be an imperative, and also helps to remove the bias of experts through its sequential treatment of pooled interview data. The scenario outcome is very different from just ranking the most "important" barriers to RE as a number, in which unyielding and often broad contextual factors would be a significant component (See Figure 3, and barriers in significant but certain quadrant, for example lack of capacity, small economy, and remote location).

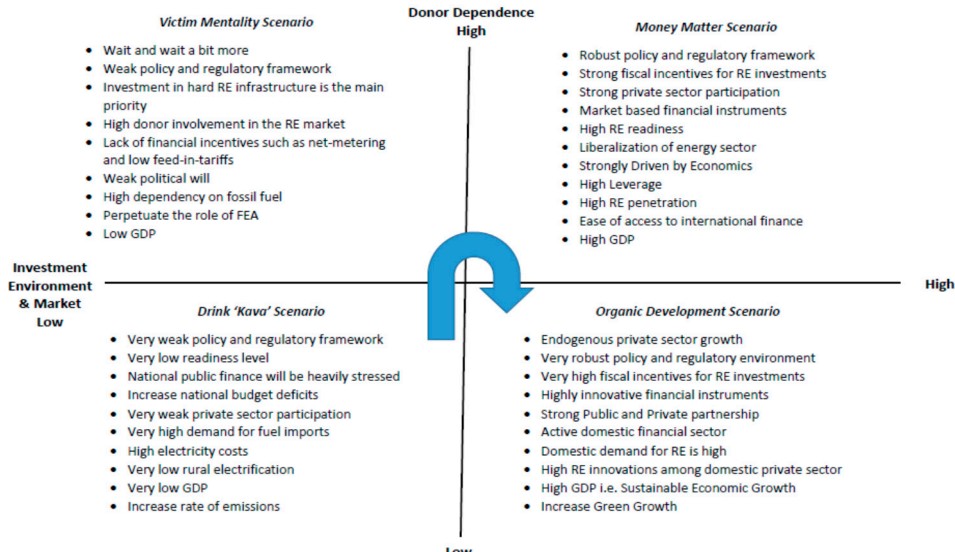

**Figure 3.** The four possible future scenarios regarding the resourcing of Fiji's nationally determined contributions (NDC).

*3.2. Scenario Validation*

Once the scenarios were developed (Figure 3), they were circulated again to the group of experts for reactions and comments. This step is critical, as it ensures that the scenarios being presented gain a sufficient level of acceptance from the expert community for the purpose of initiating a strategic conversation amongst the key stakeholders on how Fiji's NDC could be sustainably resourced. The buy-in from key stakeholders provides assurance that the results presented in this paper can contribute to the overall discussion on how Fiji could successfully achieve its energy target.

## 4. Results

*4.1. Step by Step Outcomes of the Five-Step Approach*

4.1.1. Plotting the Barriers by Significance and Uncertainty

Figure 1 below illustrates the plotting of the common barriers of RE investment in Fiji on to the axes of "significance" and "uncertainty". Figure 1 is the cumulative results of Step 1, 2 and 3.

4.1.2. New Emerging Axes

Figure 2 below illustrates the two new emergent cluster areas from the plotting of the barriers. These two new axes spanned low to high donor dependence, and low to high quality of investment environment and market. Figure 2 is the result of Step 4.

4.1.3. Developing the Scenarios

Figure 3 below illustrates the four possible future investment scenarios that reflects the combined significant but uncertain barriers as per the new emergent axes. Figure 3 details possible scenarios that are based on the four quadrants, which have varying levels of donor dependence and quality of investment and market. Figure 3 is the result of Step 5.

*4.2. Narrative of the Future Investment Scenarios*

The "drink kava scenario" is derived from a social and leisure situation common in the Fijian culture and in most PICs, where a group of people will idly sit and drink kava—a narcotic sedative drink made from the crushed roots of a native shrub—just to pass time. It is closely associated with a typical Fijian "carefree attitude" in relation to how they view uncertainty. This future scenario posts a situation where the availability of financial resources will be very limited because of the decreasing support from donors and the domestic private sector. The burden of financing the NDC will ultimately fall on the GoF, and given the past trend of the GoF spending on social and economic priorities such as education, health, and infrastructure, those priorities are more likely to supersede that of Fiji's commitments to the NDC. Under the drink kava scenario, the likelihood of Fiji achieving its energy target is very slim.

The victim mentality scenario presents a future situation that, to a larger extent, mirrors the current RE investment climate in Fiji. As per this scenario, there is both a general lack of appetite from the domestic private sector and the GoF to commit significant resources for investment in RE, shifting such investment responsibilities, instead, to donors. The unique and special circumstances of PICs as well as their "moral privilege" of being low emission contributors, while being at the front line of victims of climate change are the main drivers for such posture. Emotional diplomacy—the strategic deployment of emotional behavior by state actors to shape the perception of others [28]—is expected to play a pervasive role in soliciting external public climate finance towards the implementation of the NDC, and there is an expectation that Fiji will exploit their moral standing in the climate change domain, as well as their extreme vulnerability to convince donors to accelerate and upscale their investments in RE.

The money matters scenario represents a future situation where Fiji's private sector can effectively catalyze RE investments from external sources. A vibrant and robust "RE investment environment" is essential for such a scenario to eventuate, and will be the main funding target of external public finance. The money matter scenario exemplifies a future where the domestic private sector is "comfortable" with investment in RE; that is, most investment barriers are eliminated, and there is a high degree of certainty about the fiscal viability of RE as an investment option.

The organic development scenario depicts a future in which there is a very high degree of domestic private sector involvement in RE investments. This scenario represents a situation where a RE-based market actually exists in Fiji. The organic development scenario also represents a more advanced level of RE investment environment, where the domestic private sector is empowered to drive the market for RE production and consumption. It also underscores a future where more of the RE value chain is driven by the domestic private sector. In this future scenario, the aim is not only to find the right RE fit for Fiji, but also the way through which the domestic private sector will be able to manufacture RE technologies, and subsequently generate more green jobs in Fiji. It is important to note that, in the Fiji context, a good example of an industry that has managed to achieve a high level of endogenous private sector growth, is the tourism sector.

At a glance, the scenario analysis presents the four future scenarios as separate and independent, on the basis of the "quadrant" assumptions that they fall in. However, when closely examined, the four future scenarios suggest a possible transition pathway that Fiji could pursue to endogenously grow its domestic private sector investment in RE (Figure 3, see blue arrow).

The outcome of the scenario analysis (i.e., Figure 3) however, only outlines a broader vision and the transition stages (future scenarios) that Fiji might go through, in order to endogenously grow its domestic private sector and deepen its participation in the RE sector. Missing from this broader picture are the resourcing "specs" in terms of what Fiji need to target so that the desired financing future scenario is attained. Based on the scenario results (Figure 3), this paper proposes a resource mobilization framework, which suggests what the funding/resourcing priorities should be in order for Fiji to reach the envisioned RE investment future. This resource mobilization framework is explained in detail in the discussion section (see Figure 4).

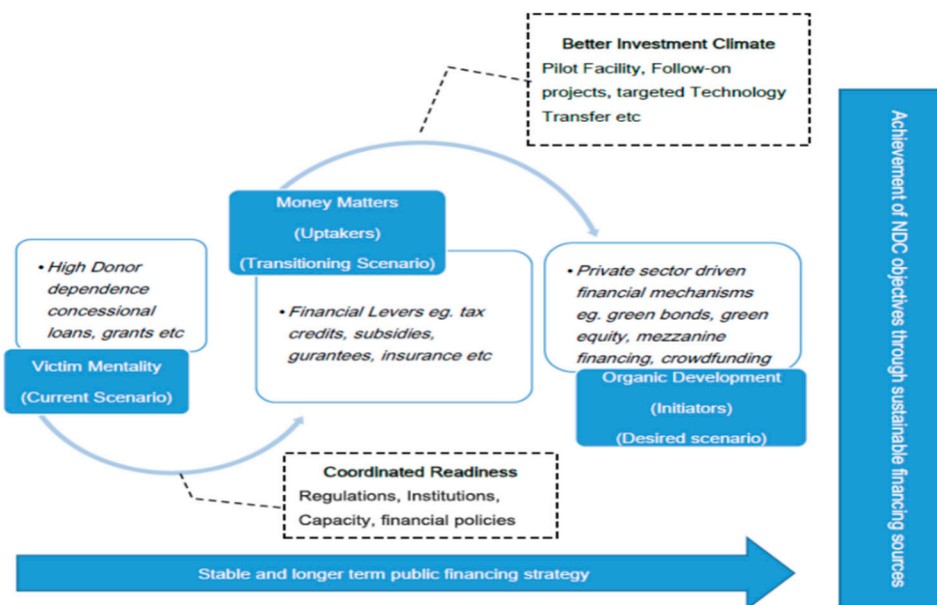

**Figure 4.** Proposed Fiji's NDC resource mobilization framework for Endogenous Domestic Private Sector Growth in the RE Sector.

## 5. Discussion

### 5.1. Current Approach towards Private Sector Investment in Fiji's Energy Sector

Fiji has undertaken a number of key reforms of its energy sector in order to attract more private investment in the sector [19]. Apart from the various energy policy reviews, regulations, and energy acts that have been enacted, the state entity that sources the majority of electricity in Fiji—Energy Fiji Ltd (EFL)—has also been privatized. Moreover, financial levers designed to strengthen the investment environment for the private sector have been introduced (Table 1). These investments incentives target not only foreign investments, but also the domestic private sector. Michalena argued that, given the current level of incentives to investors, Fiji's RE investment environment is one of the most subsidized in the world [29].

**Table 1.** Business opportunities to investment in Fiji's energy sector [30]. IPP—independent power producer; RE—renewable energy.

| Investment Opportunity | Incentives |
|---|---|
| 1. IPP Tariff Rate * <br> * applies to all RE investments | • 33.08 cents VEP (VAT exclusive price) |
| 2. Bio-Fuel | • 10-year tax holiday for new activity but minimum level <br> • Duty free importation of assets required to establish the factory <br> • Duty free on chemicals for bio-fuel production <br> * To qualify, investors total investment must be >1 million FJD and must employ >20 people |
| 3. Renewable Energy Production and Power Co-Generation | • Five years of tax holidays for new activity |
| 4. Energy Efficient Equipment | • Five years of tax incentives (only VAT paid) for imported equipment |
| 5. RE Equipment | • Five years of tax incentives (only VAT paid) for imported equipment |
| 6. Foreign Investment | • No minimum investment needed for investment in energy sector |

Financial policies have also been introduced, targeting the use of instruments that are designed to attract domestic private investments in RE. Examples include the directive to all commercial banks in Fiji to ring-fence 2% of their lending portfolio to RE projects [31], and the setting up of the Sustainable Energy Development Facility by the Fiji Development Bank (FDB), which provides cheaper financing terms to domestic private investors who plan to invest in RE technologies [32]. In the build up to the Conference of the Parties (COP) 23, Fiji issued a sovereign green bond, which raised 50 million USD from private sources [33]. Grants, loans, and equity are the three main financial instruments being used to raise new finance in RE domestically, and it is estimated that between 2014 and 2017, these instruments contributed to 119 million USD worth of investments in Fiji's energy sector [32]. Fiji plans to extent the use of these financial instruments to include new and innovative financial instruments, in order to attract more domestic private investments in the electricity sector.

Past financing trends to Fiji indicate that the country is one of the largest recipients of RE-related assistance in the Pacific, because it has been endowed with a wide source of natural RE [17,23]. The RE investment portfolio in Fiji is, however, largely geared towards hydro-power generation. RE projects and infrastructure in the country are primarily financed by external public finance [29]. Reasons for dependency on external assistance is due to the capital-intensive nature of RE technologies, and the inability of the GoF and the domestic private sector to fully fund large scale RE projects [17,29].

The continued reliance on public finance to fund large scale RE projects continues to crowd out private sector investment in the sector, because it creates minimal financial incentives to seriously pursue such endeavors [18,29]. Fiji's private sector is generally reluctant to investment in RE projects, because of the perception that investments are driven by external parties, and the perceived risks associated with RE investments [29]. There is a danger that if the current RE financing prioritization persists, the uptake of RE in Fiji will lag further behind the global trend, and as a consequence, both its energy security aspirations as well as their NDC target may not be achieved [29].

*5.2. The Resource Mobilization Framework—A Proposition from the Scenario Analysis*

The outcomes of scenarios suggest that there is a strong need for donors and the GoF to re-orient their current funding priorities and strategies in order to achieve NDC. More importantly, the specific resourcing priorities must be approached with a long-term perspective. Illustrating this resourcing pathway is critical to both the GoF and its donors, because it highlights the areas where they need to channel and concentrate their public climate finance in order to propel the Fijian private sector towards a future where it can create and sustain the market for RE. Therefore, as per the proposed resource mobilization framework (Figure 4), the desired future RE investment scenario that Fiji should aspire to, is the organic development scenario.

The organic development scenario is directly aligned with the 2014 Fiji's Green Growth Framework and the 2017 National Development Plan, which have acknowledged the need for more domestic private sector participation in financing Fiji's sustainable development pathway. The increased involvement of the domestic private sector, especially in the energy sector, tends to create innovative green employment opportunities, build capacities for expansions into other green areas, and can also provide co-benefits across the spectrum of the Sustainable Development Goals (SDGs), such as poverty reduction, health and wellbeing, education, economic growth and so on. More importantly, the organic development scenario will directly contribute to the achievement of SDGs 7 and 13, which revolve around the aim of affordable and clean energy and climate actions. Achieving this future RE investment state will require finance to be channeled in a targeted manner, with a long-term perspective of strengthening specific areas in the RE investment environment.

A critical assessment of Fiji's 2030 NDC Implementation Road Map indicates that the GoF plan is to maintain the investment strategy, which emphasizes external public finance channeled towards "large and hard" RE projects, in order to achieve its NDC target. The current investment strategy is thus synonymous with the victim mentality scenario. The proposed set of actions advanced by the NDC Implementation Road Map emphasizes more investments in concrete emission reduction projects, such as the installations of more solar photovoltaic systems, biomass, waste to energy plants, and hydro plants. Investing in these initiatives is necessary, as they are aligned with the general objectives of the NDC in reducing Fiji's global emission contributions. However, there are questions as to whether pursuing the same resource strategy of utilizing limited public finance to fund RE infrastructural projects will result in an achievement of the NDC target, as experts have continuously argued that such financing modality on its own is not sufficient to cover the cost of the investments that are needed [17–19,21,23,29,34,35].

To break away from the victim mentality scenario, the GoF and donors must undertake concerted efforts to channel their resources towards the money matters scenario, where the underlying crux is the internal mobilization of domestic private finance. Readiness is the critical link between these two scenarios, and thus, should be the main target of funding. Readiness as per this paper is understood as the creation of the investment environment that will attract and stimulate domestic private sector investments, rather than the narrow definition advanced by the Green Climate Fund and the Adaptation Fund, which tend to emphasize the direct access of climate finance from specific sources. To attract private finance in the energy sector, donors and the GoF should re-orient the funding priorities from investment in technically building RE projects, to supporting and strengthening initiatives that remove barriers for domestic private investments in the energy sector.

The enhancement of the energy sector governance arrangements through the strengthening of the regulatory/policy frameworks, institutional capabilities, capacity building, and financial policies are readiness activities that are critical in removing investment barriers in the energy sector. Efforts to strengthen Fiji's RE investment environment have been actively pursued by the government, although complex chains of entrenchment are still apparent [36]. Dornan has argued that the regulatory reform undertaken by Fiji in the energy sector serves as an ideal model for PICs, because it is domestically driven rather than from donor pressure [18]. Fiji has been able to make significant gains in strengthening its RE investment environment through the establishment of an independent regulator, which has managed to increase electricity tariffs, opening the opportunity for domestic private sector investment to flow [18]. The current efforts being pursued by the GoF and its donors to "ready" the RE investment environment for domestic private investments signals that the shift from the victim mentality scenario towards that of a money matter scenario is underway.

However, the continuous lack of domestic private sector investment in RE, despite Fiji's "advanced" readiness progress, indicates that major gaps still exist regarding its readiness approach. Jafar argued that the major reason that RE continuously fails to become a viable investment option in Fiji is because donors prefer to fund short term RE technical initiatives, rather than providing stable long-term funding for domestic private sector development in RE [37]. While Dornan observed that donors in the Pacific are slowly moving towards program-based RE assistance, and away from the project-based modality [34], Betzold found that investment in the "hardware component" (i.e., equipment, infrastructure, and distribution) still accounts for the bulk of finance of such programs [35]. The continuous emphasis on investing in "hard" RE projects rather than in the strengthening of the domestic private sector role, tends to negate the gains made in "readying" Fiji's RE investment environment, because it crowds out the domestic private sector from the RE investment space.

The crowding out effect is reflected in the high level of uncertainty and perception of risks that Fiji's domestic private sector associate with RE investments. Such an unfavorable outlook of RE investments, despite the market maturity of some RE technologies, is common among domestic financial institutions. There is a need to extend Fiji's current readiness from just focusing on the reforms of the energy sector, to also considering the strengthening of the role of domestic financial institutions in RE investments. Efforts to strengthen the participation of Fiji's financial institutions in RE investments have largely been ad hoc, and mainly limited to short term workshops.

There is also a need for donors to support more long-term programs that specifically target the domestic financial institutions' role in RE investments. The Sustainable Energy Financing Project (SEFP), which is supported by the World Bank in partnership with the Australia and New Zealand Banking Group (ANZ) and the FDB, was designed to increase the uptake of RE in Fiji by guaranteeing 50% of participating banks' RE related lending through the World Bank's risk-mitigation facility [38]. The lessons learnt from the SEFP are vital, and should be used by donors as the basis of mobilizing resources to support and design similar initiatives that will target the remaining domestic private sector participants who did not benefit from the SEFP [38].

There is also a need to provide stable and long-term funding to initiatives that will allow the domestic private sector to better absorb the financial and technical risks associated with RE investments. Examples include partial guarantees for RE lending, concessional credit lines, and staff secondment with international institutions, such as the International Finance Corporation. These initiatives have been proven to be successful with the domestic private sector of other developing countries [39]. Therefore, the readiness approach in Fiji must not only focus on attracting domestic private investments, it must also involve long-term support for initiatives that strengthen the domestic private sector's capacity and experience in the RE sector.

While important, readiness is just a transitional state towards unlocking the full potential of Fiji's domestic private finance in RE. Readiness as envisioned in the money matter scenario represents a future where Fiji's domestic private sector has become confident with the idea of RE as a mainstream investment option, and is willing to mobilize finance towards RE uptake.

However, for private finance to become a sustainable source of RE investments, the domestic private sector should be transformed from being mere up-takers and initiators of RE, to drivers of RE investments. For this scenario to occur, the domestic private sector must endogenously grow and develop the energy sector—the organic development scenario.

Innovation is a critical ingredient for endogenous domestic private sector growth. While there are realistic limitations on the ability of Fiji's private sector to be serious innovators in terms of RE technologies because of their small economies, the right amount of support could potentially lead to developing new financing modalities and financial packages designed to support sustainable RE development. A good example of such financial innovation in PICs is the Secured Transaction Framework, a financing mechanism that makes it easier for lenders to accept movable assets such as vehicles, inventory, account receivables, and even crops as collateral for loans [40]. So far, more than 50,000 new loans under this scheme have been granted by financial institutions [40], and potential on how such initiatives could be translated into investments for RE should be explored.

Pilot RE projects have also been argued to be an essential enabler for innovation in the domestic private sector [17]. Pilot projects, when successful, not only enhance market familiarity with new technologies, but also advance RE towards commercialization (i.e., up-scaling). While the success of pilot RE projects in Fiji has been a mixed bag [41], it has also been observed that there is a lack of uptake in cases where RE projects have been successful [42]. The lack of innovative RE technology adoption by the domestic RE private sector can be attributed to the ad-hoc following up of projects building and resourcing, rather than coming up with innovative schemes. The financing of successful pilot projects in Fiji are largely "once off" in nature, with little commitments from donors to channel long term resources towards replicating such success in other local communities or inventing more appropriate RE settings. The channeling of resources towards innovation is a critical initiative in the process of creating a much better RE investment environment, as this would not only contribute to the growth of RE investments, but would also promote the endogenous growth of RE through creativity, while at the same time properly nesting social and financial benefits for communities. Initiatives that strengthen targeted technology transfers in developing countries can lead to the development of new business areas that also involve the introduction of innovative technologies that are relevant to the local context [43].

The innovation concept applies to innovative financial mechanisms too. Confidence in RE will mobilize and unlock the full potential of private sector RE investments. Such confidence will not only be manifested in the new RE technologies that will be introduced in the market, but also in the willingness to adopt innovative financial instruments introduced by the government. The domestic private sector needs to drive these innovative financial mechanisms so as to transform the Fiji's electricity sector, and to also ensure a sustainable resourcing pathway for Fiji's transition to a low-carbon economy in the long run.

A targeted approach towards promoting endogenous domestic private sector growth is also very relevant to the newly created Regional Pacific NDC Hub. The hub is a regional platform that consists of technical experts (long term and short term) that will deliver demand driven technical assistance to PICs for the implementation of their NDC. The hub provides an ideal platform where Fiji and PICs can consolidate their technical know-how (i.e., local and international) and act as a clearing house for their RE technical issues. Taibi has argued that the ability to locally create knowledge on RE technologies is essential in promoting a "paradigm shift" in the investment behavior for domestic private sectors; thus, shifting away from an assistance base, towards self-sustaining large scale deployment of RE in-country [44].

## 6. Conclusions

Fiji's NDC has outlined an ambitious target to transform its energy sector by 2030. While many have hailed such an ambition as courageous, the resourcing of such initiatives is a cause of concern. An investment totaling 2.97 billion USD is required, of which 54% is conditional on Fiji receiving

significant means of implementation and support from other sources. Considering the major climate finance windfall and the high degree of uncertainty of climate finance availability that currently exists in the international climate finance architecture, the "billion-dollar question" therefore relates to how Fiji would attract sustainable funding to implement its NDC. With private finance being identified as the recourse for such a shortfall to fully unlock its potential, the GoF and its donors need to strategically channel limited public finance in a sustained manner, which will mobilize domestic private finance in the long run.

Despite Fiji's donors consistently prioritizing investments in RE infrastructures, there are indications that they are starting to move towards funding incentives designed to attract domestic private sector investments in RE. Donors are now supporting the strengthening of the investment environment by helping developing countries like Fiji to implement an array of readiness initiatives. Readiness is critical in removing investment barriers in RE, however, is not sufficient to facilitate long-term domestic private sector investments in RE. Readiness initiatives are mainly designed to enable the domestic private sector to adopt RE technologies. For the domestic private sector to be agents of achieving the envisioned change of the NDC, the private investors must become RE initiators. To be an initiator will require innovations, and for the domestic private sectors to assume this status, they must be allowed to endogenously grow and develop Fiji's RE market.

Using the scenario analysis technique, this paper formulated a resource mobilization framework, which outlined important initiatives that donors and the GoF could target in order to endogenously grow the private sector. Sustained financing for innovations, follow-on projects from successful pilot projects, and targeted technology transfers are critical to the organic growth of Fiji's domestic private sector. This paper argues that donors and the GoF should significantly re-orient their NDC funding priorities, and commit long-term resources towards these approaches, to transform the role of the domestic private sectors to be the primary drivers of the RE sector in Fiji.

In the absence of a refocus on priorities on how Fiji's NDC is to be resourced, there is a risk that not only will the energy targets be missed (again), but that the overall sustainable development path currently being pursued might be unattainable. Leveraging the full potential of domestic private investment is critical for accelerating and sustaining climate change efforts in the long run, and provides many co-benefits in terms of "green" jobs and securing wellbeing. Without genuine efforts to channel external public climate finance towards endogenously growing the domestic private sector, the NDC runs the risk of joining a growing list of "feel good" national initiatives that bear very few real benefits to local communities.

**Author Contributions:** The paper was conceptualized by J.S. and J.H., J.S. wrote and compiled the paper, conducted the data collection and analysis, and prepared the table and figures. Professor J.H. and E.M. both assisted in supervising the development of the paper and closely assisted with the compilation of the manuscript, application of the method, data analysis and the editing. This paper is derived from J.S.'s Thesis which was co-supervised by Professor J.H.

**Funding:** This research received no external funding.

**Acknowledgments:** The authors would like to thank Mathew Dornan for his critical insights in the first draft of this paper. We also acknowledge the constructive comments received from the two blind reviewers of this journal, which provided guidance for further refinement of the paper. This paper is a streamlined version of the fourth Chapter of Jale Samuwai's PhD Thesis submitted to The University of South Pacific in 2018, titled *Will the Tide of Climate Finance Change Finally Turn in Our Favour? Three Essays on Accessing and Mobilizing Climate Finance in Oceania Post Paris Agreement.*

**Conflicts of Interest:** The authors declare no conflict of interest.

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
