# Peer review of "Thinking Outside the Box: Deepening Private Sector Investments in Fiji’s Nationally Determined Contributions through Scenario Analysis"

_sustainability, doi:10.3390/su11154161_

Round 1
Reviewer 1 Report
- Please check all abbreviations are explained before use - e.g. "VEP" in Table 1.
- When citing at the beginning of a sentence, please use the name of main author or institution: e.g. line 403
- By any chance, please shorten the discussion - this section is very long
- Please deepen the sections on RE investments, public finances, and domestic finance within the Introduction. I would be good to have a look to Europe, as it is one of the largest economies worldwide providing mentioned funding. Among others, please consider:
https://www.mdpi.com/1996-1073/9/12/990
http://publications.jrc.ec.europa.eu/repository/handle/JRC95364
https://ec.europa.eu/clima/policies/adaptation/financing/funds_en
Minor issues:
- Please fix the track change at line 8
- Line 12: Please write "more than" instead of using the symbol ">"
- Line 33 etc.: Please correctly indicate degree Celsius "°C"
- Line 55 etc.: Please correctly use the "-" sign
- Line 274: Projects/infrastructure - Please see also line 503
- References: please indicate the links in a proper way (without "<")
Author Response
We note with appreciation the comments received by the Reviewer. Please see in attachment our detail response as per the comments received and we hope that we have sufficiently addressed the comments received.

Reviewer 2 Report
This manuscript examines Fiji’s renewable energy (RE) goals as outlined in the country’s nationally determined contribution (NDC) to the United Nations Framework Convention on Climate Change (UNFCCC). Specifically, the manuscript discusses how the investments necessary to achieve Fiji’s RE goals can be mobilised, and to what extent Fiji’s domestic sector can provide the necessary investments.
The manuscript has three distinct parts: it first introduces the Fijian economic and energy context and presents challenges and barriers to the RE transition in Fiji (and many other developing countries), such as a strong reliance on external donor funding, weak regulatory frameworks, and small domestic markets. This part also notes some steps that the Government of Fiji as well as its donor partners have already undertaken, or plan to undertake, to address some of these challenges, including tax incentives for RE investments or a tendency to provide funding through (longer-term) programmes rather than (short-term) projects.
Part II then introduces the scenario technique, and explains how the authors identified RE barriers, classified them as (in)significant and (un)certain, and developed four scenarios based on the significant and uncertain barriers. Part III finally builds on the four scenarios from Part II to discuss where Fiji currently stands and suggest ways to reach the most desirable “organic development” scenario.
The manuscript is rather comprehensive and discusses energy policy, RE barriers and investment scenarios in Fiji and other SIDS. This comprehensiveness, however, comes at the expense of a clear research question and red thread. My main comment thus is to re-think the paper’s structure and to find ways to streamlining the manuscript. By better connecting the three parts, the authors would also improve the text flow and make their argument and contribution(s) more visible and explicit. I list a few suggestions on how the authors could achieve this.
First, a clear research question right at the beginning of the paper would be very helpful. Is the question on p. 4, why RE investments are not forthcoming, the key research question? If so, put this right upfront, and re-focus the manuscript on this question. At the moment, the paper touches upon so many different things that it is at times difficult to follow. For example, the role of the scenarios approach remains somewhat unclear, and I was a bit surprised to start reading about the methods on p. 9, well into the manuscript. I would have expected a clearer separation of the literature review, methods, and results/discussion section (at least for an empirical paper). An overview of the entire paper and its structure at the end of the introduction would be helpful.
Second, I don’t think that section 2 really explains the scope of the study. Rather, it presents information on the Fijian energy sector – although this information is somewhat scattered. Section 2.2 is extremely short and only contain Fiji’s energy goals, including transforming its electricity grid to be 100% renewable. It would be more useful to discuss Fiji’s electricity mix, the extent of its grid and the difficulties of powering remote islands, rather than present data on Fiji’s GDP composition. The latter information is not essential to the paper’s argument, I think. In contrast, more information on the energy sector would be useful, and should be concentrated in one place. The authors note that Fiji currently relies on 53% hydro, 45.5% diesel/fuel, 0.39% wind and 1.1% Independent Power Producers (IPPs). Do these IPPs all include solar, biomass and waste-to-energy mentioned on 6? How many people (% of the population) are covered by Fiji’s energy grid? How many people have in fact access to electricity? How does the situation compare to the PSIDS overall?
Third, Fiji already seems to be doing quite a bit with regard to RE, as described in section 2.4. Table 1 for example lists several tax incentives, though it is unclear why these actions are presented only in the form of a table rather than included in the main text. Additionally, section 2.4 discusses Fiji’s future plans to obtain RE finance. There are also actions by donors to shift their RE investments to more capacity building, longer-term programmes, and the like. Why is this information discussed before the scenarios are introduced? Wouldn’t it make more sense to discuss Fiji’s and donors’ current and future activities when discussing the resource mobilisation framework? This would also allow the authors to highlight where they see Fiji “on track” and where Fiji’s actions are misaligned with their framework.
Fourth, I think that the scenario approach is quite interesting, but the role of this analysis right now is unclear. Is the paper an empirical interview-based study? If so, the authors should spend much more time explaining their method, the interviews they conducted, and the input from the experts they talked to. Or is the paper more a discussion of Fiji’s energy policies, barriers, and suggestions for mobilising RE investments? If so, then what is the added value and role of the interviews? It is also not quite clear to me how the authors derived the 50 barriers, how they ordered the barriers in figure 1, and how they then came to the barriers and ordering in figure 2.
Overall, I think that the paper has a contribution to make, but the authors will need to thoroughly revise the manuscript. I think that these revisions need to focus not so much on the paper’s substance itself, but on its presentation. A clearer red thread and argument would help; in particular, with a clear research question, the authors could streamline their paper, be much more concise and efficient, and improve the text flow.
Moreover, the paper needs to be carefully proof-read for language and style. There are a lot of typos, grammar mistakes, and very long sentences. The authors should avoid the passive voice as much as possible.
Author Response
We note with appreciation the constructive comments received. They were really useful in further streamlining and the readibility of the presentation of the paper. Our detail response to the 4 broad concerns/suggestions of the reviewer are as attached. We hope that it has sufficiently meet the reviewers concern.

Round 2
Reviewer 1 Report
The melioration indications have been well implemented.
Author Response
Please refer to attached Adjoiner for details.

Reviewer 2 Report
This revised manuscript has somewhat improved from the original version. The paper’s first part is clearer and presents Fiji’s energy sector more efficiently than before. However, many of my original comments and concerns remain, and should be addressed before the manuscript can be considered for publication, as outlined below.
First, and as already noted in my first review, the manuscript’s structure needs to streamlined and improved. I still don’t quite understand the structure. Part 2 provides an overview of Fiji’s energy sector, which to me seems to be part of the paper’s findings rather than a literature review. This section already explains some barriers to private RE investment in Fiji (such as over-reliance on external finance) and some solutions (e.g. tax incentives in table 1). Section 3 presents the method as well as results (e.g., the four scenarios). Section 4 discusses the results, i.e., again talks about barriers to RE investments and suggests ways of overcoming these. Based on my reading, I don’t quite see how the discussion builds on or relates to the four scenarios identified in section 3. I suggest to clearly separate the methodology from the results, and to discuss barriers to RE investments (as well as potential solutions) in one section only, rather than spread this discussion all over the manuscript. If the authors had a clear research question, framed as such, the manuscript’s argument and red thread would become clearer, I think.
Second, the scenario approach is interesting but its role in the overall argument remains unclear, and the way the scenarios were derived and applied remains opaque. First, why was it necessary to conduct a literature review and identify 50 barriers (some of which appear already in sections 1 and 2)? Couldn’t you directly have asked the RE experts for the most significant and uncertain barriers? Indeed, what does “consultation” mean? Did you conduct interviews? How did you plot the barriers in figure 1? Did barriers at the top (e.g., lack of capacity, experience, knowledge) receive a higher ranking than barriers at the top (e.g., bureaucracy)? How did you draw the lines to differentiate the four quadrants? Why is it necessary to further categorise the “critical uncertainties”? To what extent are the first steps necessary to derive the four scenarios? It seems a bit like you could come up with the four scenarios based just on your knowledge of the Fijian energy sector and RE barriers?
Third, the manuscript needs to be carefully proofread and edited for style and clarity. There are typos and grammatical errors throughout. The authors also tend to use the passive voice rather than the active voice, noun constructions rather than verbal construction, and very long sentences, all of which make the text rather heavy and difficult to read. For example, on p. 2 (line 83), the authors write:
“Using the case of Fiji, a PIC, this paper seeks to answer the research question posed by this paper by exploring a potential resource mobilization strategies [sic] that could be adopted to unlock the potential of the domestic private sector to finance the NDC.”
This sentence could be made much more reader friendly, for example:
“Using the case of Fiji, this paper explores a potential resource mobilisation strategy, which could enable the domestic private sector to finance Fiji’s NDC”, or something along these lines. Examples such as this one abound throughout the manuscript.
Some minor comments:
United Nations Framework Convention on Climate Change (p. 2, l. 45)
“Private financing has been advocated as the panacea…” (p. 2, l. 64) – by whom?
Does the private sector really only account for 20% of Fiji’s gross national (or gross domestic?) product? (p. 3, ll. 101f)
I am not sure I can folllow the second justification for selecting Fiji as a case study (p. 2, ll. 105ff)
The authors write twice how they reduced the originally 50 barriers to just 25 (p. 7, ll. 265f, and ll. 284f)
Examples would be helpful to illustrate barriers from each quadrant (p. 8, ll. 303ff)
Trial and error as a method is rather unclear (p. 8, l. 316)
Is figure 2 really necessary? Does it add anything?
What were the reactions and comments of the experts to the four scenarios (p. 11, ll. 379ff)?
Can you provide a reference for the expert opinion on the inadequacy of public finance (p. 12, 431f)? Or do you here refer to the group of experts you consulted for the paper?
How realistic are the money matters and organic growth scenarios? Will Fiji reach these scenarios and fund its RE transition with private finance?
Author Response
Please refer to attached Adjoiner for comments.

Round 3
Reviewer 2 Report
Investment Scenarios for Achieving Energy Transition in Developing Countries: A Case Example from Fiji
Sustainability Manuscript # xx
This second revision has considerably improved from the original version. I find the structure more convincing. While there remains room for further improvement, I recommend to publish this version with only minor revisions. I have mostly two remaining concerns.
First, I think that the structure has improved. However, the results section now includes only three figures, whereas the discussion clearly includes both a discussion of the results as well as the actual results. The authors for example present the four scenarios in section 5.2.1 (where figure 3 would fit much better).
Second, there remain a number of typos and grammatical errors throughout the text. I list some that I found – the list is unlikely to be complete.
· l. 12: NDCs or NDC? Please check for consistency throughout.
· l. 43: (NDCs), which contain (comma and verb in plural, i.e., no –s)
· l. 93: 1.1% supplied by (remove “to be”)
· l. 98f: fuel imports accounts for (no to) 14–17% of GDP, which is relatively higher than in other PICs
· l. 105: economics, geopolitics (noun needed)
· l.124f: studies … like those (not that) of [18, 21, 23] have made (not suggested) numerous policy suggestions. (Otherwise “Suggesting suggestions”)
· l. 149: successful in the area of strategic – what? (noun missing)
· l. 164: the study/analysis used a 5-step methodology (Otherwise: methodology used a methodology)
· l. 310: capital intensive nature (not intensiveness nature)
· l. 417: tend to emphasise
· l. 437: Dornan has (not have) observed (or even better: Dornan observed)
Subject to these improvements, I recommend publication.
Author Response
Please see attached Adjoiner for details.
